# Kinetically restrained oxygen reduction to hydrogen peroxide with nearly 100% selectivity

Jinxing Chen[1,2], Qian Ma[1,2], Xiliang Zheng[1], Youxing Fang [1], Jin Wang[3✉] & Shaojun Dong [1,2✉]

Hydrogen peroxide has been synthesized mainly through the electrocatalytic and photo-catalytic oxygen reduction reaction in recent years. Herein, we synthesize a single-atom rhodium catalyst ($Rh_1$/NC) to mimic the properties of flavoenzymes for the synthesis of hydrogen peroxide under mild conditions. $Rh_1$/NC dehydrogenates various substrates and catalyzes the reduction of oxygen to hydrogen peroxide. The maximum hydrogen peroxide production rate is 0.48 mol $g_{catalyst}^{-1} h^{-1}$ in the phosphorous acid aerobic oxidation reaction. We find that the selectivity of oxygen reduction to hydrogen peroxide can reach 100%. This is because a single catalytic site of $Rh_1$/NC can only catalyze the removal of two electrons per substrate molecule; thus, the subsequent oxygen can only obtain two electrons to reduce to hydrogen peroxide through the typical two-electron pathway. Similarly, due to the restriction of substrate dehydrogenation, the hydrogen peroxide selectivity in commercial Pt/C-catalyzed enzymatic reactions can be found to reach 75%, which is 30 times higher than that in electrocatalytic oxygen reduction reactions.

[1] State Key Laboratory of Electroanalytical Chemistry, Changchun Institute of Applied Chemistry, Chinese Academy of Sciences, Changchun 130022, China. [2] University of Science and Technology of China, Hefei 230026, China. [3] Department of Chemistry and Physics, Stony Brook University, Stony Brook, NY 11794, USA. ✉email: jindwang12@163.com; dongsj@ciac.ac.cn

Hydrogen peroxide ($H_2O_2$) is a green oxidant with various uses, mainly in medical disinfection, wastewater treatment, industrial bleaching, and chemical synthesis[1]. Currently, the industrial production of $H_2O_2$ relies on the energy-consuming anthraquinone oxidation/reduction process[2,3]. In recent years, the photocatalytic and electrocatalytic oxygen reduction reaction (ORR) has received increasing attention for the direct on-site production of $H_2O_2$[4,5]. The main efforts have been focused on designing nanomaterials from the composition and structure aspects to improve the selectivity of oxygen reduction to $H_2O_2$, with representatives including carbon-based catalysts and single-atom catalysts[6-8]. In the electrochemical synthesis process, many catalysts have to work in alkaline electrolytes to reduce the overpotential of the ORR. However, the low oxidability and easy decomposition of the produced $H_2O_2$ under alkaline conditions hindered the subsequent application for oxidation reactions[9,10]. The photocatalytic method is also not suitable for the mass production of $H_2O_2$ due to the low efficiency and instability of $H_2O_2$ under illumination[3,11,12].

In contrast, $H_2O_2$ in organisms can be produced under mild conditions by reactions catalyzed by enzymes, in particular flavin-containing enzymes[13]. Flavin-containing enzymes are a large class of oxidoreductases whose active center is flavin, including flavin adenine dinucleotide (FAD) and flavin mononucleotide (FMN)[14]. Flavins are extremely versatile cofactors capable of accepting electrons from various biomolecules (electron donors) and then donating electrons to another molecule (electron acceptor). According to the differences in electron donors and electron acceptors, flavoenzymes can be divided into oxidases, reductases, and monooxygenases. When the electron acceptor is $O_2$, the enzyme is a common flavin-dependent oxidase (e.g., glucose oxidase or alcohol oxidase), which efficiently catalyzes the $O_2$ reduction reaction to produce, in most cases, $H_2O_2$. Since flavin-dependent oxidases can not only catalyze the oxidation of substrates to target products but also produce $H_2O_2$, they are widely used in biology, medicine, detection and environmental fields. Considering the high price and ease of inactivation of natural enzymes, mimicking flavin-dependent oxidases and finding substitutable applications of their natural counterparts is of great scientific and practical significance. To realize the characteristic enzymatic activities of a flavin-dependent oxidase, the catalyst should not only be able to abstract hydrogen and electrons from the substrate, that is, induce a dehydrogenation reaction but also catalyze the reduction of $O_2$ to $H_2O_2$ through the $2e^-$ pathway without energy input, such as illumination or electricity. However, most catalysts are designed for a specific half-reaction[15], either substrate oxidation or oxygen reduction, thereby failing to mimic an oxidase.

In this work, we propose a general and convenient approach to synthesize nitrogen-doped carbon-supported single-atom rhodium (Rh), iridium (Ir), and cobalt (Co) catalysts by using melted urea as a solvent and nitrogen source. The single-atom Rh catalyst shows the highest ability to catalyze the reduction of $O_2$ to $H_2O_2$ in the presence of glucose, alcohols, amines, formic acid, NADH, or phosphorous acid and thus exhibits flavin-dependent oxidase-like activities. We found that the kinetics of the electrocatalytic oxygen reduction are very different from those of the enzyme-like oxygen reduction. In the electrocatalytic ORR, electrons are continuously transferred from the working electrode with a lower applied potential to the catalyst for the ORR. The $H_2O_2$ selectivity depends on the difference in the thermodynamic stability in the catalytic oxygen reduction process caused by the intrinsic properties of the electrocatalysts. In the enzymatic ORR, $O_2$ reduction is only activated under the premise of catalytic dehydrogenation of the substrate, while a single catalytic site of catalysts can only catalyze the removal of 2 electrons per substrate molecule.

Therefore, the subsequent $O_2$ reduction is kinetically restrained to $H_2O_2$ through the typical two-electron pathway with very high selectivity.

## Results

**Synthesis and characterization of the Rh₁/NC catalyst.** Figure 1a illustrates the simple preparation process for the $Rh_1$/NC catalysts. In the first step, a urea solid was melted into a liquid at 150 °C, and then, poly-(ethylene glycol) (PEG) and $RhCl_3$ were added to the urea liquid (Supplementary Fig. 1). A uniform mixed solution was formed after stirring for 5 min. In the second step, the mixed solution was poured into a crucible and pyrolyzed at 900 °C for 2 h under a $N_2$ atmosphere. During pyrolysis, urea was thermally polymerized into $g$-$C_3N_4$, while PEG was transformed into amorphous carbon at temperatures below 650 °C. The N atoms in the $g$-$C_3N_4$ layer can anchor and prevent the aggregation of Rh atoms (Supplementary Fig. 2). Upon further increasing the temperature, $g$-$C_3N_4$ was decomposed and volatilized, accompanied by Rh, N, and C doping on the carbon substrate formed by the carbonization of PEG to obtain the N-doped carbon-supported single-atom Rh catalyst ($Rh_1$/NC). Although urea contains a large amount of elemental carbon, an additional carbon source (i.e., PEG) is still needed in the synthesis process. This is because the $g$-$C_3N_4$ produced by the urea thermal polymerization will completely volatilize when the temperature is higher than 650 °C (Supplementary Fig. 3).

Scanning electron microscopy (SEM) images show that the synthesized $Rh_1$/NC catalyst had a wrinkled nanosheet shape (Fig. 1b). No obvious Rh nanoparticles were detected in the high-resolution transmission electron microscopy (HRTEM) images (Fig. 1c). High-angle annular dark-field scanning transmission electron microscopy (HAADF-STEM) coupled with in situ energy-dispersive X-ray spectroscopy (EDS) elemental mapping images showed that the Rh, N, and C elements were uniformly distributed across the nanosheets (Fig. 1d and Supplementary Figures 4, 5) with a loading content of 0.89 wt% (Supplementary Table 1). From the aberration-corrected HAADF-STEM images, atomically distributed Rh atoms were clearly identified as highly isolated bright dots because of the higher Z-contrast (Fig. 1e, f).

To further investigate the atomic dispersion and coordination environment of Rh species in $Rh_1$/NC, X-ray absorption fine structure spectroscopy (XAFS) tests were conducted (Fig. 1g and Supplementary Fig. 6). The Fourier transformed extended X-ray absorption fine structure (FT-EXAFS) spectra of the Rh K-edge showed an intense peak at 1.5 Å, corresponding to the Rh−N coordination shell. The single peak representing the Rh−Rh shell ca. 2.3 Å (compared with Rh foil) was not observed, further indicating the atomic dispersion of Rh atoms in $Rh_1$/NC catalysts[16]. FT-EXAFS fitting of $Rh_1$/NC (Fig. 1g and Supplementary Table 2) based on the DFT model demonstrates that each Rh atom is coordinated with four N atoms. Based on the high solubility of metal salts in melted urea, our synthesis strategy can be applied to synthesize a family of atomically dispersed metal catalysts, including $Ir_1$/NC and $Co_1$/NC (Supplementary Figure 7). The aberration-corrected HAADF-STEM images demonstrated atomically dispersed Ir or Co atoms throughout the samples (Fig. 2 and Supplementary Figures 8–11). The FT-EXAFS spectra of $Ir_1$/NC and $Co_1$/NC further confirmed the exclusive presence of atomic metal atoms (Fig. 2)[17], as only the Ir−N or Co-N coordination shell signal was detected (Supplementary Figures 12 and 13).

In the absence of urea, melted PEG molecules can also serve as a solvent to disperse metal salts. Thermal decomposition of the precursor containing only PEG and Rh(acac)₃ resulted in carbon-supported Rh nanoparticles (Rh/C) (Fig. 2g). TEM images and

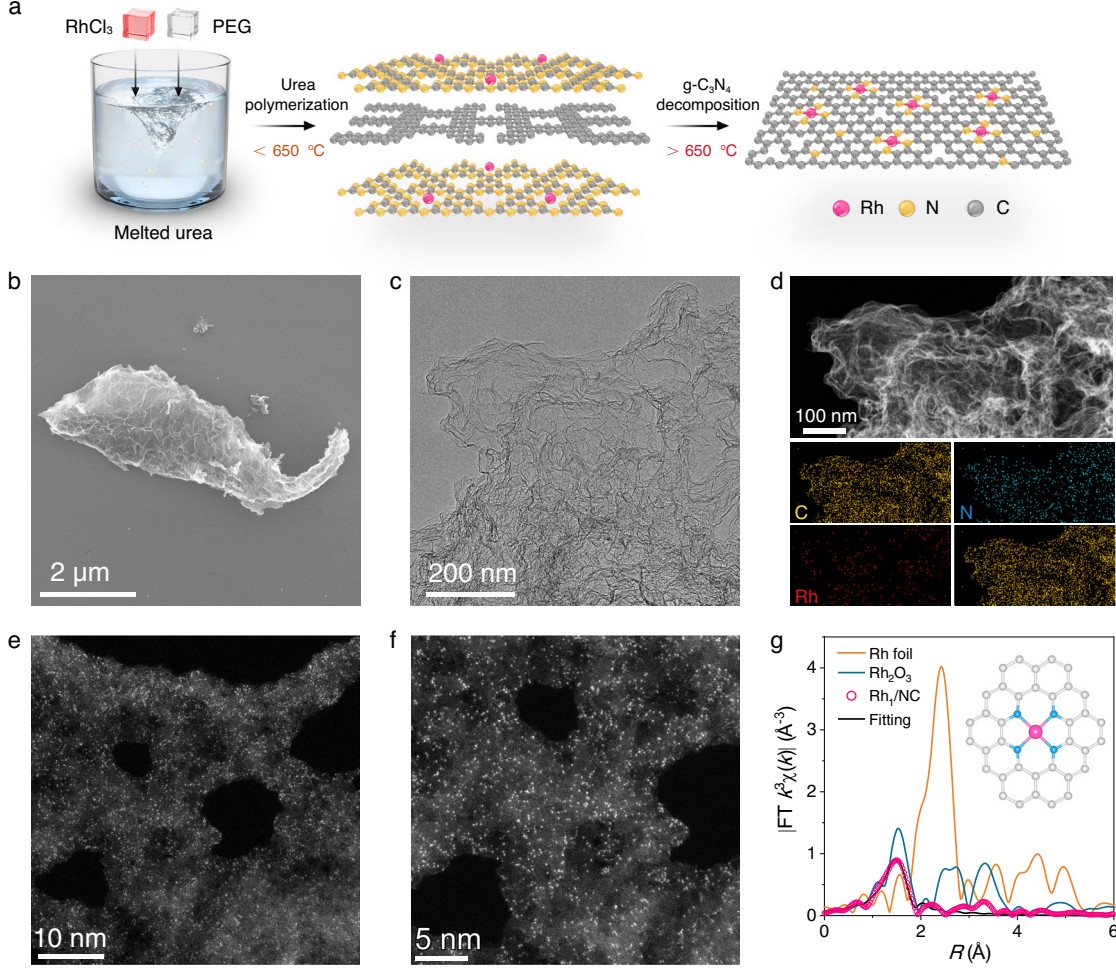

**Fig. 1 Synthesis and characterization of Rh₁/NC. a** Schematic of the synthesis process of Rh₁/NC. TEM **b** and SEM **c** images of Rh₁/NC. **d** HAADF-STEM image and the corresponding energy-dispersive X-ray elemental mapping of Rh₁/NC. **e, f** Atomic-resolution HAADF-STEM images of Rh₁/NC. **g** FT-EXAFS spectra at the Rh K-edges of Rh₁/NC Rh foil and Rh₂O₃. Insert: structure of RhN₄ for EXAFS fitting and DFT calculation. Experiment (Fig. 1b–f) was repeated 3 times independently with similar results.

XRD spectra showed the existence of metallic Rh nanoparticles, and this result can in turn prove the crucial role of urea in promoting the generation of atomically dispersed and homogeneous Rh–Nx moieties in Rh₁/NC. The powder XRD pattern of Rh/C showed characteristic carbon and metallic Rh diffraction peaks, which implied the presence of Rh nanoparticles (Fig. 2h and Supplementary Fig. 14). Brunauer–Emmett–Teller (BET) analyses showed that Rh₁/NC possessed a high surface area and a mesoporous structure compared to Rh/C (Fig. 2i). This is because a large amount of g-C₃N₄ produced by urea thermal polymerization can facilitate the dispersion of PEG. Raman spectroscopy confirmed the formation of graphitized carbon in Rh/C and Rh₁/NC (Supplementary Fig. 15). The chemical states of Rh₁/NC and Rh/C were analysed by X-ray photoelectron spectroscopy (XPS). The high-resolution XPS spectra indicated that the metal in the single-atom catalysts was positively charged, in agreement with the XANES spectra (Supplementary Figures 16 and 17).

**Oxidase-mimicking activities of metal single-atom catalysts.**
Natural flavoenzymes can catalyze the dehydrogenation of various substrates, accompanied by the reduction of O₂ to H₂O₂ (Fig. 3a). Flavoenzyme catalysis obeys the Ping-Pong mechanism containing two individual half-reactions: substrate dehydrogenation and O₂ reduction[18,19]. Flavoenzymes also have analogous

catalytic sites: the His residue acts as a Brønsted base to promote the dissociation of H protons, and then FAD transfers electrons and protons from different substrates to O₂ to produce H₂O₂ (Fig. 3b)[20]. Here, aerobic oxidation of benzyl alcohol was chosen as a model reaction to evaluate the enzyme-mimicking properties of the as-obtained Rh₁/NC (Supplementary Figures 18 and 19). After the reaction of Rh₁/NC and benzyl alcohol for 10 min, H₂O₂ was produced in the mixture solution. The H₂O₂ production rate was accelerated with increasing pH (Fig. 3c). This is because OH⁻ can act as a Brønsted base to promote the dissociation of hydroxyl groups in benzyl alcohol, similar to the imidazole group in the active center of flavoenzymes[21]. Rh₁/NC can transfer electrons from benzyl alcohol not only to O₂ but also to other artificial electron acceptors, such as [Fe(CN)₆]³⁺ (Supplementary Fig. 20)[22]. This finding indicates that the benzyl alcohol oxidation process does not rely on reactive oxygen species. The O₂ only acts as an electron acceptor, which is consistent with the Ping-Pong reaction, to scavenge the electrons and protons produced by the dehydrogenation step and recover the catalyst site of Rh₁/NC to its initial state.

The alcohol dehydrogenation and oxygen reduction steps in the aerobic oxidation of benzyl alcohol can be studied on the electrode (Fig. 3d)[23]. The onset potential of Rh₁/NC for benzyl alcohol oxidation was ~0.5 V_RHE (RHE, reversible hydrogen electrode) in the anodic sweep. For oxygen reduction, the onset

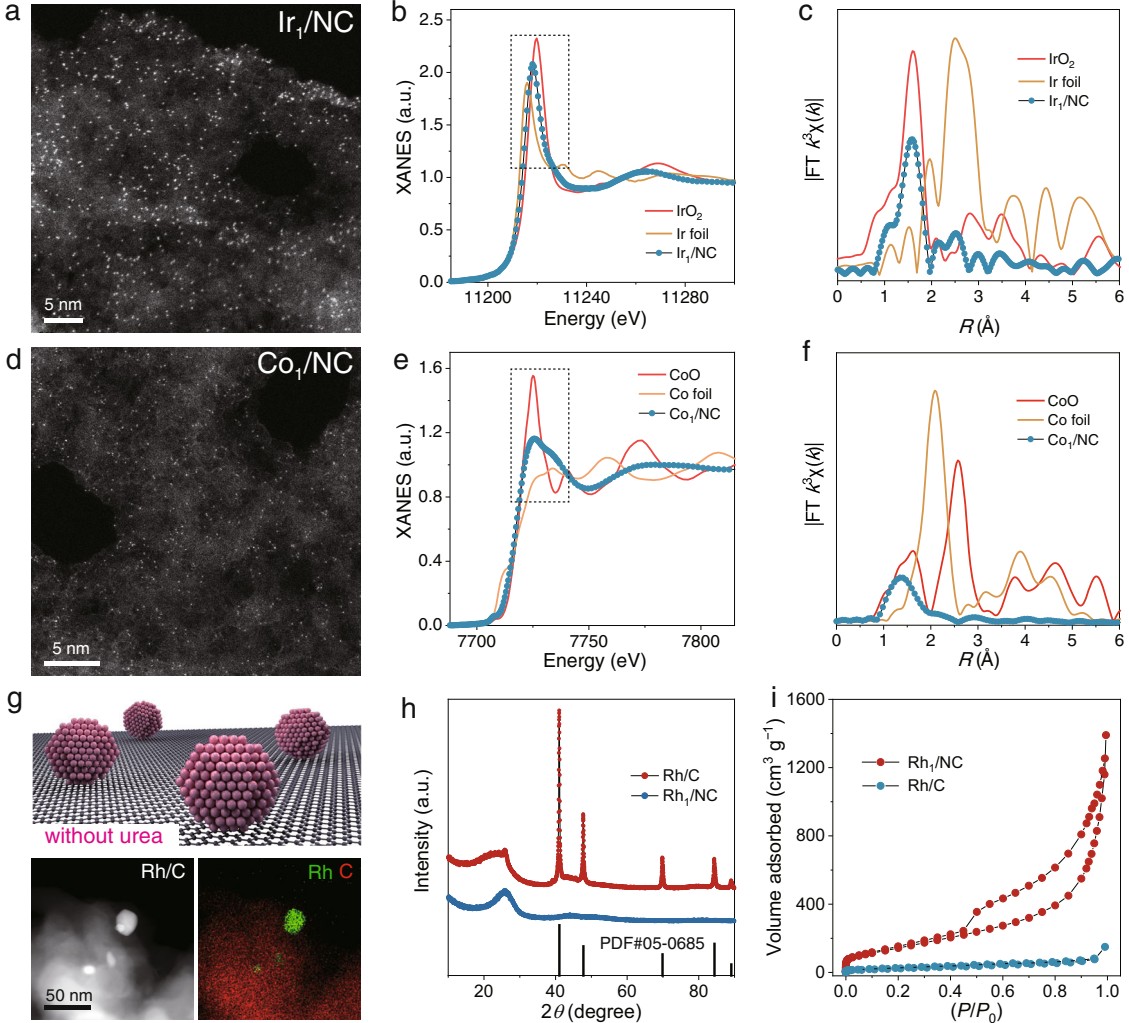

**Fig. 2 Characterization of Ir₁/NC, Co₁/NC and Rh/C. a** Atomic-resolution HAADF-STEM images of Ir$_1$/NC. **b** Ir L-edge XANES spectra of Ir$_1$/NC, Ir foil, and IrO$_2$. **c** FT-EXAFS spectra at the Ir L-edges of Ir$_1$/NC, Ir foil, and IrO$_2$. **d** Atomic-resolution HAADF-STEM image of Co$_1$/NC. **e** Co K-edge XANES spectra of Co$_1$/NC, Co foil, and CoO. **f** FT-EXAFS spectra at the Co K-edges of Co$_1$/NC, Co foil, and CoO. **g** Schematic illustration of Rh/C synthesized without urea and the HAADF-STEM image with energy-dispersive X-ray elemental mapping of Rh/C. **h** XRD patterns of Rh$_1$/NC and Rh/C. **i** N$_2$ adsorption-desorption isotherms of Rh$_1$/NC and Rh/C. Experiment (Fig. 2a, d, g) was repeated 3 times independently with similar results.

potential was ~0.85 V in the cathodic sweep. The potential difference (0.35 V) between the cathode and anode indicates that Rh$_1$/NC can spontaneously catalyze the aerobic oxidation of benzyl alcohol, which is consistent with the experimental results. The catalytic activity of Ir$_1$/NC and Co$_1$/NC for the oxidation of benzyl alcohol to H$_2$O$_2$ was lower than that of Rh$_1$/NC (Fig. 3e). The potential difference and current intensity are closely related to the catalytic activity of the alcohol aerobic oxidation reaction (Supplementary Fig. 21)[24]. H$_2$O$_2$ was also detected in the process of Au-catalyzed oxidation of benzyl alcohol, while the content of produced H$_2$O$_2$ was much lower than that for the single-atom catalyst, and no H$_2$O$_2$ was observed in Pt/C-catalyzed oxidation of benzyl alcohol. In addition to catalytic oxidation of benzyl alcohol, Rh$_1$/NC can catalyze the dehydrogenation of various substrates to produce H$_2$O$_2$ in the presence of O$_2$ (Fig. 3f and Supplementary Figures 22–25).

To rapidly generate H$_2$O$_2$, HCOOH and H$_3$PO$_3$ were selected as electron donors. We first determined the kinetic parameters by varying the concentration of the substrates. The initial reaction rate ($V_0$) was well fitted to the standard Michaelis−Menten equation[25], indicating that the catalytic kinetics of Rh$_1$/NC match those of natural enzymes (Fig. 4a). The lower $K_M$ value for H$_3$PO$_3$

compared to HCOOH suggested that H$_3$PO$_3$ has a higher binding affinity to Rh$_1$/NC. The catalytic constant ($k_{cat}$, calculated based on the molar concentration of the Rh element) of H$_3$PO$_3$ was higher than that of HCOOH, showing better H$_2$O$_2$ production efficiency in Rh$_1$/NC-catalyzed H$_3$PO$_3$ aerobic oxidation. The maximum mass activity in terms of the H$_2$O$_2$ production rate was 0.48 mol g$_{catalyst}^{-1}$ h$^{-1}$. The catalytic efficiency of the single-atom catalysts is on the same order of magnitude as that of electrocatalysts and 3 orders of magnitude higher than that of photocatalysts (Supplementary Table 3–5). Of note, the H$_2$O$_2$ production rate reached a maximum when the pH was approximately 3 (Fig. 4b). Different from benzyl alcohol, the relative formation rate of H$_2$O$_2$ decreased with a further increase in pH. This is because the p$K$a values of HCOOH and H$_3$PO$_3$ are very low, indicating that HCOOH and H$_3$PO$_3$ can be dissociated in large quantities under acidic conditions (Supplementary Fig. 26). When the pH is lower than 3.5, increasing the pH contributes to the dissociation of the O-H bond in the −OH group, so the reaction rate increases. When the pH is higher than 3.5, a large amount of HCOOH and H$_3$PO$_3$ dissociate, so the activity will not be further improved. In contrast, excessive HCOO$^-$ ions, H$_2$PO$_3^-$ ions and OH$^-$ ions will be strongly adsorbed on the catalyst site, resulting in a decrease in

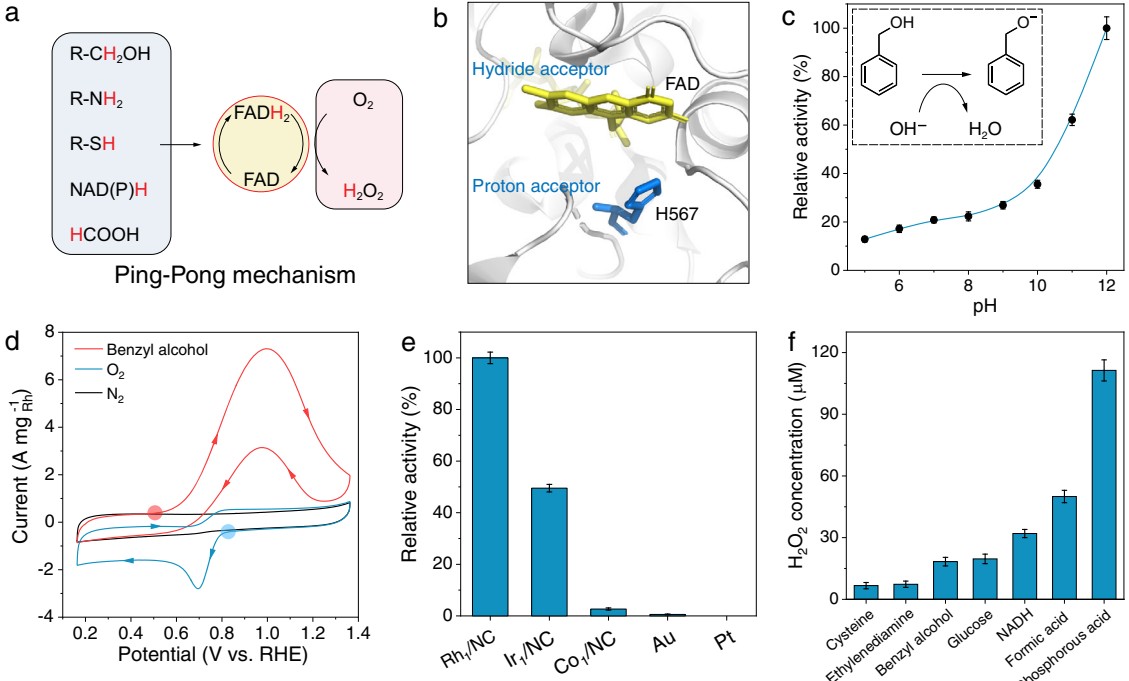

**Fig. 3 Oxidase-mimicking activities of metal single-atom catalysts. a** Schematic of flavoenzymes catalyzing the dehydrogenation of various substrates and $O_2$ reduction to $H_2O_2$. **b** Active site of the alcohol oxidase from *Pichia pastoris*. **c** pH-dependent $H_2O_2$ production rate in the $Rh_1$/NC-catalyzed benzyl alcohol oxidation reaction. **d** CV curves of $Rh_1$/NC in a $N_2$-saturated 0.1 M NaOH + 50 mM benzyl alcohol solution and in $N_2$-saturated and $O_2$-saturated 0.1 M NaOH solutions with a scan rate of 100 mV s$^{-1}$. **e** Relative $H_2O_2$ production rate in the benzyl alcohol oxidation reaction with different catalysts. **f** $H_2O_2$ concentrations after the reaction of $Rh_1$/NC (20 µg mL$^{-1}$) and different substrates (10 mM, 0.1 mM for NADH and cysteine) for 5 min. Data are presented as mean values (SD). (The error bar represents the standard deviation of 3 independent measurements).

activity[26]. Notably, the main purpose of synthesizing $H_2O_2$ is to exert its oxidizability in specific oxidation reactions. Thus, an acidic pH is not only beneficial to the preservation of $H_2O_2$ but also facilitates $H_2O_2$ participation in the oxidation reactions according to the Nernst equation[27].

Since $H_3PO_3$ has buffering capacity, the synthesis of $H_2O_2$ can be directly carried out in a mixed solution of $H_3PO_3$ and potassium phosphite ($KH_2PO_3$). $Rh_1$/NC, $Ir_1$/NC, and $Co_1$/NC can catalyze the oxidation of phosphite to continuously produce $H_2O_2$ with the flow of $O_2$. A 50 µg mL$^{-1}$ amount of $Rh_1$/NC produced 1 mM $H_2O_2$ within 6 min (Fig. 4c). The commercial Pt/C catalyst (3 nm Pt nanoparticles supported on carbon) with the same metal content as the single-atom catalysts can only produce a very small amount of $H_2O_2$ under the same test conditions. The lower $H_2O_2$ production was initially thought to be due to the low $H_2O_2$ selectivity of the Pt-catalyzed ORR. However, a large amount of $H_2O_2$ can be produced with increasing Pt/C dosage (Supplementary Fig. 27), implying that the $H_2O_2$ selectivity in the enzymatic reaction is higher than that in the electrocatalytic ORR.

DFT calculations were carried out to gain insight into the catalytic activity. The process of HCOOH dehydrogenation is divided into two elementary reactions: the breakage of the O-H bond in the -OH group ($HCOOH \rightarrow HCOO + H$) and the C-H bond rupture ($HCOO \rightarrow CO_2 + H$). The $H_3PO_3$ oxidation follows a similar reaction path: −OH group dissociation ($H_3PO_3$ to $H_2PO_3$) and P-H bond rupture ($H_2PO_3$ to $HPO_3$) (Supplementary Figures 28 and 29). According to the free energy profiles, the $H_3PO_3$ oxidation on $Rh_1$/NC is more accessible with the lower energy barriers for both dehydrogenations step (Fig. 4d). The calculated activity order is consistent with the experimental results. The first step of −OH group dissociation is the rate-limiting step with a quite high energy barrier for both HCOOH and $H_3PO_3$. In terms of the energy barrier, the overall catalytic performance of

$Rh_1$/NC is very low. This is because the effect of OH$^-$ in solution on the reaction was not considered in the theoretical simulation. Therefore, the calculated results correspond to the catalytic performances under highly acidic conditions. Under practical reaction conditions, most HCOOH and $H_3PO_3$ have already dissociated to HCOO$^-$ and $H_2PO_3^-$ due to the low pKa value (3.7 for HCOOH, 1.3 for $H_3PO_3$). Thus, $Rh_1$/NC only needs to catalyze the breaking of C-H in HCOO$^-$ or P-H in $H_2PO_3^-$ with a relatively low energy barrier (Fig. 4d)[26,28,29].

The charge transfer and proton release caused by oxidative dehydrogenation are prerequisites for the subsequent ORR. The metal atom provides its electrons for the adsorption to coordinate, and it also provides an empty d orbital to gain electrons. In the course of the dehydrogenation of formic acid, the Hirshfeld-I charge of Rh is initially 1.158e and changes to 1.169e, 1.614e, 1.625e, 1.367e and 1.551e in the next S1, TS1, S2, TS2, and S3 structures, respectively (Fig. 4e). The charge density difference analysis of $Rh_1$/NC also showed an obvious electron accumulation on Rh after adsorption with $2H^*$ (Fig. 4f). On the basis of DFT calculations, the Rh atom acquires electrons in the whole HCOOH and $H_3PO_3$ oxidative dehydrogenation process. The dehydrogenation reaction transfers the electrons from the substrate to the catalysts for subsequent $O_2$ reduction[30,31].

**$H_2O_2$ selectivity in the enzymatic and electrocatalytic ORRs.** Commercial Pt/C can produce a large amount of $H_2O_2$ in $H_3PO_3$ aerobic oxidation, which is contrary to the well-known low $H_2O_2$ selectivity in the electrocatalytic ORR. This interesting phenomenon leads us to speculate that the process of the enzymatic ORR is different from that of the electrocatalytic ORR. In the electrocatalytic ORR, electrons are continuously transferred from the working electrode with a lower applied potential to the catalyst for the ORR (Fig. 5a). The supply of electrons is not the

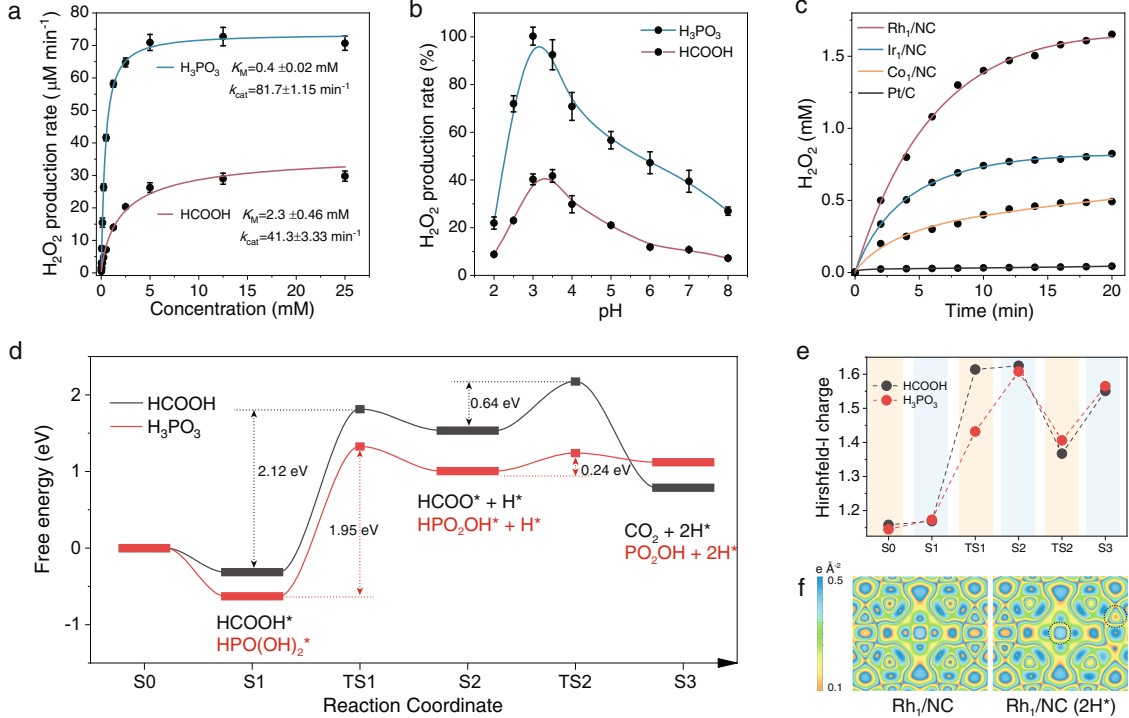

**Fig. 4 Rh₁/NC catalyzes HCOOH and H₂PO₃ oxidation for H₂O₂ production. a** Initial $H_2O_2$ production rates ($V_O$) in Rh₁/NC (10 μg mL$^{-1}$)-catalyzed HCOOH and H₃PO₃ oxidation reactions. The $V_O$ values were fitted to the standard Michaelis−Menten equation. **b** pH-dependent $H_2O_2$ production rate in the Rh₁/NC-catalyzed HCOOH and H₃PO₃ oxidation reaction. **c** Time-dependent $H_2O_2$ concentration in 2.5 mM H₃PO₃ + 2.5 mM KH₂PO₃ in the presence of different catalysts (50 μg mL$^{-1}$ for single-atom catalysts, 5 μg mL$^{-1}$ for Pt/C). **d** Free energy profiles for oxidative dehydrogenation of HCOOH and H₃PO₃ catalyzed by Rh₁/NC through the formate and phosphite pathway (without the assistance of a Brønsted base). **e** Hirshfeld-I charge of Rh at different reaction coordinates. **f** Charge density difference of Rh₁/NC and Rh₁/NC adsorbed with 2H. The inserted black circle indicates the adsorption site of H. Data (Fig. 4a, b) are presented as mean values (SD). (The error bar represents the standard deviation of 3 independent measurements).

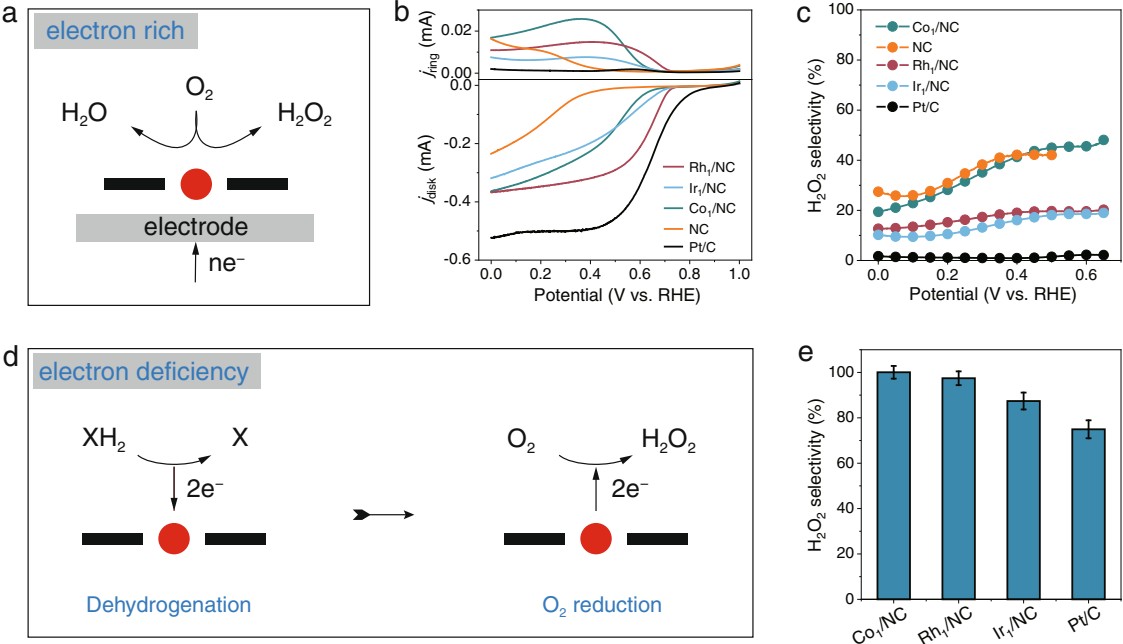

**Fig. 5 H₂O₂ selectivity in the enzymatic and electrocatalytic ORR. a** Schematic of the electrocatalytic ORR catalyzed by single-atom catalysts. **b** Polarization curves at 1600 r.p.m. and simultaneous $H_2O_2$ detection currents at the ring electrode in 0.1 M acetate buffer + 0.1 M KCl (pH=4). **c** Calculated $H_2O_2$ selectivity at various potentials. **d** Schematic of the dehydrogenation of substrates and the ORR process catalyzed by single-atom catalysts. **e** $H_2O_2$ selectivity in NADH aerobic oxidation in 0.1 M acetate buffer (pH = 4). Data (Fig. 5e) are presented as mean values (SD). (The error bar represents the standard deviation of 3 independent measurements).

restriction of the reaction. In this case, the selectivity of $O_2$ reduction to $H_2O_2$ caused by the intrinsic properties of different catalysts must be studied. The electrocatalytic ORR was performed on a rotating ring-disc electrode (RRDE). The $H_2O_2$ produced on the disc electrode diffused to the ring electrode and was oxidized at a fixed potential of 1.2 V. Typically, the polarization curve of Pt/C showed a much higher onset potential and current, as well as a remarkably lower ring current (Fig. 5b and Supplementary Figure 30). The $H_2O_2$ selectivity of Pt/C was calculated to be ~2% from the ring current and disc current (Fig. 5c). $Rh_1/NC$, $Ir_1/NC$ and $Co_1/NC$ exhibited higher $H_2O_2$ selectivity (~15% for $Rh_1/NC$ and $Ir_1/NC$, 40% for $Co_1/NC$ in the potential range of 0–0.6 V) with respect to commercial Pt/C.

In the enzymatic ORR, the electrons needed for oxygen reduction are provided by the substrate, while a single catalytic site of the catalysts can only catalyze the removal of 2 electrons per substrate molecule. Therefore, the subsequent $O_2$ can only obtain 2 electrons to reduce to $H_2O_2$ through the typical two-electron pathway (Fig. 5d). The supply of the electrons is limited by the dehydrogenation of the substrate. Thus, $O_2$ has difficulty carrying out the typical 4-electron reduction process. Theoretically, the selectivity of $O_2$ reduction to $H_2O_2$ can reach 100%. Here, we used NADH aerobic oxidation as a model reaction to evaluate the $H_2O_2$ selectivity. Because NADH oxidation is accompanied by a decrease in the characteristic absorption value at 340 nm, it is helpful for accurately determining substrate consumption[32]. The $H_2O_2$ selectivities for $Rh_1/NC$, $Ir_1/NC$, and $Co_1/NC$ were all higher than 90% (~100% for $Co_1/NC$, and $Rh_1/NC$) (Fig. 5e). For Pt/C, the $H_2O_2$ selectivity was (~75%) 30 times higher than that in the electrocatalytic ORR. The selectivity for Pt/C did not reach 100% of the theoretical value, which may be due to the side reaction of $H_2O_2$ decomposition (Supplementary Figures 31 and 32). Since the decomposition of $H_2O_2$ is significant with increasing pH[33], we measured the $H_2O_2$ selectivity at pH 7. As expected, the $H_2O_2$ selectivity of all catalysts was reduced (Supplementary Fig. 33), implying that the side reaction will lead to the detected selectivity being lower than the real value. Therefore, in addition to improving the selectivity of $O_2$ reduction to $H_2O_2$, avoiding the occurrence of side reactions is also worthy of attention.

## Discussion

In this study, we proposed a universal method to synthesize single-atom catalysts using melted urea as the solvent. Urea is used not only as a solvent to dissolve and disperse Rh and PEG but also as an N source doped on the carbon substrate to form an N-coordinated single-atom catalyst. The experimental and theoretical results demonstrated that single-atom catalysts can effectively catalyze the oxidation of various substrates to a target product and produce $H_2O_2$. $H_2O_2$ production can occur under acidic conditions, which is beneficial to the preservation of $H_2O_2$ and facilitates the participation of $H_2O_2$ in oxidation reactions. We discovered that the enzymatic ORR process is in a state of electron deficiency and thus induces very high $H_2O_2$ selectivity compared to the electrocatalytic ORR. Due to the restriction of substrate dehydrogenation, the $H_2O_2$ selectivity in commercial Pt/C-catalyzed enzymatic reactions can reach 75%, which is 30 times higher than that in electrocatalytic $O_2$ reduction reactions. These results imply that kinetic restrictions are more effective in improving the $H_2O_2$ selectivity than regulating the intrinsic properties of catalysts.

## Methods

**Materials**. Urea, 3,3′,5,5′-tetramethylbenzidine (TMB), and poly(ethylene glycol) (PEG 4000) were purchased from Aladdin. $RhCl_3$, $IrCl_3$, $CoCl_2$, and Nafion solutions were purchased from Aldrich. NADH was purchased from Genview.

Horseradish peroxidase (HRP, 300 U·mg$^{-1}$) was purchased from Roche. Pt/C (20%) was purchased from Alfa Aesar. Deionized (DI) water was used in all our experiments.

**Synthesis of $Rh_1/NC$, $Ir_1/NC$, and $Co_1/NC$**. In a typical procedure, 5 g urea was added to a vial (volume, 20 mL) and heated at 150 °C to melt it into a transparent liquid, followed by the addition of 200 mg PEG 4000 under stirring. PEG 4000 was dissolved in liquid urea within 1 min to form a clear solution. Then, 0.005 mmol $RhCl_3$ was added and stirred for 5 min. The mixture solution was poured into an alumina crucible, pyrolyzed in a tube furnace under a $N_2$ atmosphere at 900 °C for 2 h with a heating rate of 2 °C min$^{-1}$, and then naturally cooled to room temperature. $Rh_1/NC$ was obtained without any posttreatment. Before heating, the tubular furnace was filled with nitrogen for 30 min to expel oxygen.

$Ir_1/NC$ and $Co_1/NC$ were prepared using a similar procedure to that of $Rh_1/NC$, except that the metal salt was changed to $IrCl_3$ (0.005 mmol) or $CoCl_2$ (0.005 mmol).

**Synthesis of Rh/C**. In a typical procedure, 2 g PEG 4000 was added to a vial (volume, 20 mL) and heated at 100 °C to melt it into a transparent liquid. Then, 0.1 mmol $Rh(acac)_3$ was added and stirred for 5 min. The mixture solution was poured into an alumina crucible, pyrolyzed in a tube furnace under a $N_2$ atmosphere at 900 °C for 2 h with a heating rate of 2 °C min$^{-1}$, and then naturally cooled to room temperature. Rh/C was obtained without any posttreatment. Before heating, the tubular furnace was filled with nitrogen for 30 min to expel oxygen.

**Characterization**. High-angle annular dark-field scanning transmission electron microscopy (HAADF-STEM) and corresponding energy-dispersive X-ray spectroscopy (EDX) analysis were performed on a Talos F200X microscope. Aberration-corrected HAADF-STEM images were obtained on a Titan Themis Z Cs-corrected scanning/transmission electron microscope (Thermo Scientific). Scanning electron microscopy (SEM) images were obtained with a ZEISS Sigma 300 field-emission microscope with an accelerating voltage of 5.0 kV. Powder X-ray diffraction (XRD) patterns were obtained with a Bruker D8 Advance instrument with Cu Kα radiation (λ = 1.54056 Å) at a scan speed of 2° min$^{-1}$. The surface elemental composition and bonding configuration of the as-prepared samples were analysed by X-ray photoelectron spectroscopy (XPS) (K-Alpha™, Thermo Scientific). The metal contents in the samples were determined by inductively coupled plasma-atomic emission spectrometry (ICP-AES). The contents of N and C atoms in the samples were determined by an Elementar Vario EL cube. $N_2$ adsorption-desorption isotherms were recorded at 77 K with a Micromeritics TriStar II 3020 analyser. The specific surface area was calculated by Brunauer–Emmett–Teller (BET) models. Raman spectra were recorded with a customized LabRAM HR800 confocal Raman microscope (Horiba Jobin Yvon).

**$H_2O_2$ production and quantification**. For $H_2O_2$ production, 20 μL $Rh_1/NC$ (0.5 mg mL$^{-1}$, dispersed by ultrasound) and 20 μL substrates (125 mM $H_3PO_4$ + 125 mM $KH_2PO_3$) were added to a vial containing 920 μL. After the solution was mixed, the reaction was carried out without stirring for several minutes at room temperature. The detailed reaction conditions (concentration and reaction time) of different experiments were added to the corresponding figure notes.

For $H_2O_2$ quantification, after the reaction of catalysts and different substrates, 20 μL HRP (0.1 mg mL$^{-1}$) and 20 μL TMB (20 mM in DMSO:EtOH = 1:9) were added to the above solution (960 μL). UV–vis absorption measurements were performed within 2 min. The produced $H_2O_2$ was quantified by a standard curve, which was drawn from a series of known concentrations of $H_2O_2$.

**$H_2O_2$ selectivity in the NADH aerobic oxidation reaction**. NADH and different catalysts were sequentially added into a vial containing 870 μL 0.1 M acetate buffer (pH = 4). The final concentrations of NADH and different catalysts were 50 μM and 40 mg mL$^{-1}$, respectively. After complete oxidation of NADH (determined by the absorbance at 340 nm), TMB and HRP were added and reacted for 1 min. The $H_2O_2$ concentration was detected based on the typical absorbance of TMBox at 650 nm. The $H_2O_2$ selectivity was calculated by the following equations:

$$H_2O_2(selectivity) = \frac{c_{H_2O_2}}{c_{H_2O}/2 + c_{H_2O_2}} \tag{1}$$

$$c_{H_2O} = (100 - 2c_{H_2O_2})/2 \tag{2}$$

where 100 is the total electron number provided by 50 μM NADH and $c_{H_2O_2}$ is the concentration of $H_2O_2$.

Therefore:

$$H_2O_2 (\%) = 400 * c_{H_2O_2}/(100 + 2c_{H_2O_2}) \tag{3}$$

**$H_2O_2$ selectivity in the electrocatalytic oxygen reduction reaction (ORR)**. For the ORR, 5 μL of the catalyst ink was dropped onto the polished working electrode of a rotating ring-disk electrode (the diameter of the working electrode was 4 mm).

The inner diameter of the ring electrode is 5 mm, and the outer diameter is 7 mm, followed by drying in air. Acetate buffer (0.1 M, pH = 4) containing 0.1 mol L$^{-1}$ KCl was used as the electrolyte. Before the electrochemical test, the electrolyte solutions were purged with O$_2$ for at least 30 min. CV and linear scanning voltammetry (LSV) curves were collected at a scan rate of 100 mV s$^{-1}$. The ORR polarization curves were collected at a rotating speed of 1600 r.p.m. The peroxide yields (H$_2$O$_2$%) were calculated by the following equations:

$$H_2O_2 (\%) = 200 * I_r / (I_r + NI_d) \qquad (4)$$

The electron transfer number ($n$) was calculated by the equation:

$$n = 4I_d / (I_d + I_r/N) \qquad (5)$$

where $I_d$ is the disk current, $I_r$ is the ring current, and $N = 0.44$ is the collection efficiency of the Pt ring.

**Density functional theory (DFT) calculations**. The first-principles calculations are performed by the Vienna Ab initio Simulation Package (VASP) with the projector augmented wave (PAW) method. The exchange functional was treated using the Perdew-Burke-Ernzerhof (PBE) functional in combination with the DFT-D correction. The cut-off energy of the plane-wave basis is set at 520 eV. All calculations, including geometry optimization, single-point energy and electronic density, were carried out within a 12.5049 × 12.2718 × 15.0000 Å$^3$ box under periodic boundary conditions, and Brillouin zone integration was performed with 2*2*1 $\Gamma$-centered k-point sampling. The self-consistent calculations apply a convergence energy threshold of 10$^{-6}$ eV. The equilibrium geometries and lattice constants are optimized with maximum stress on each atom within 0.02 eV/Å. In the Z direction, there is an approximately 14 Å vacuum for erasing the effect of periodic conditions for the slab model. The free energy ΔG can be calculated as follows:

$$\triangle G = \triangle E + (ZPE) - T\triangle S \qquad (6)$$

The specific tool Vaspkit was used for the free energy correction in calculating several ΔG values.

**Reporting summary**. Further information on research design is available in the Nature Research Reporting Summary linked to this article.

## Data availability
The data that support the findings of this study are available from the corresponding author upon request.

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

## Acknowledgements
J.C., Q.M., Y.F., and S.D. thank the support from the National Natural Science Foundation of China (No. 22074137, S.D.) and the Ministry of Science and Technology of China (Nos. 2016YFA0203203 and 2019YFA0709202, S.D.). We thank Prof. Ying Wang for help with DFT simulation.

## Author contributions
J.C. designed the studies, prepared the samples, performed the catalytic tests, and wrote the paper under the direction of the project. X.Z. helped with DFT analysis. Q.M. and Y.F. helped with the data analysis. J.W. and S.D. supervised the project and established the final version of the paper. All authors discussed the results and commented on the manuscript.

## Competing interests
The authors declare no competing interests.
