## [Peer Review File · Nature Communications]

REVIEWER COMMENTS

Reviewer #1 (Remarks to the Author):

This work synthesized a single-atom Rh catalyst (Rh1/NC) to mimic the properties of flavoenzymes for the synthesis of H₂O₂ under mild conditions. The results indicated that Rh1/NC can dehydrogenate various substrates and catalyse the reduction of O₂ to H₂O₂. The maximum H₂O₂ production rate is 0.48 mol g_{catalyst}⁻¹ h⁻¹ in the H₃PO₃ aerobic oxidation reaction. The results are very interesting, the manuscript is well-organized and easier to follow. I recommend it to be published in NC after a minor revision.

1. In the Fig. 3c (in the case of benzyl alcohol solution), the activity increases with the increasing of the solution pH, however, in Fig. 4b (in the case of H₃PO₃ and HCOOH), the activity decreases with the increasing of the solution pH when the solution pH higher than 4. More discussion is needed.
2. Line 213, It may be clearer if “-OH group dissociation” is revised to “the breakage of the O-H bond in -OH group”.
3. line 217, the “(Figure 3d)” should be “(Figure 4d)”?
4. The Pt in the commercial Pt/C is Pt single atom or Pt nanoparticles? If it is Pt particles, the size can be indicated.

Reviewer #2 (Remarks to the Author):

In this manuscript, a single-atom rhodium catalyst (Rh1/NC) to catalyse the reduction of O₂ to H₂O₂ in the phosphorous acid aerobic oxidation reaction, the maximum H₂O₂ production rate is 0.48 mol g_{catalyst}⁻¹h⁻¹ and the selectivity of O₂ reduction to H₂O₂ can reach 100%, is reported. Although there is some depth in the review, especially in the characterization of single-atom rhodium materials. However, according to previous studies, precious metal Rh-based material can catalyze O₂ reduction to H₂O₂. In this paper, C₃N₄ is used as N-doped carbon-supported to prevent the aggregation of Rh atoms, thus improving the efficiency of rhodium catalyzed of O₂ reduction to H₂O₂. So how does metal Rh-C₃N₄ carrier interact with each other? Does another nitrogen-rich porous carbon carrier material have better catalytic efficiency? It is worth further investigation. Additionally, considering the highlight or innovation of this manuscript is also not attractive. I think this paper is insufficient to be published in Nature Communications. Several recommendations are listed as follows:

1. C₃N₄ was synthesized by carbonization of urea and PEG at high temperature. How was the structure of C₃N₄ determined and what was the yield? The content of Rh in RH-NC materials needs to be revealed by more accurate characterization
2. Then the interaction between metal Rh-C₃N₄ carrier needs to be revealed. What is the advantage of using C₃N₄? During the synthesis of Rh1/NC, “the mixture solution was... 2 °C/min”, whether the process is safe.
3. What does the standard card correspond to that should be shown in the the XRD patterns (Figure 2(g)). The size dimension of all SEM or TEM images should be visualized. The tables used should be presented in a three-line format. Since the rotating ring-disk electrode of N=0.44 is used in this paper, the size of its needs to be clear.

4. Ru/NC synthesized by urea and PEG with RuCl₃ at other ratios and different temperatures should also be explored.

5. As the format, the typeface in this manuscript is not unified, such as Page 11, line 325 and Page 12, line 335, and the references need to be carefully formatted, such as: 5, 5, 9, 10, 11.... Please check the PDF file before submitting your manuscript.

Reviewer #3 (Remarks to the Author):

The authors present an interesting report into a flavoenzyme-mimicking single atom catalyst for H₂O₂ production. The presentation of a distinct electron transfer mechanism versus traditional electrocatalysts is convincing and well supported by computational and experimental data. It is interesting to see the same concept being applied to multiple metals and substrates within the same report as well. I have a couple of minor comments, mostly focused on how this catalyst could be employed for wider scale production of H₂O₂. Other than these I am happy to recommend this article for publication.

I may have missed this, but I couldn't see a reference to the experimental setup for the enzymatic H₂O₂ production. Factors such as mass loading of Rh/NC, dispersion method, if it was stirred during reactions etc.

Have the authors considered how the Rh/NC could be used for lab-scale or industrial-scale H₂O₂ production? Electrocatalysts have the advantage of existing reactor designs, and flow-by electrochemical cells can remove the H₂O₂ as it is produced into separate storage containers. Presumably the Rh/NC would need a separate process of H₂O₂ extraction and also Rh/NC recovery for batch production?

Similarly, the authors show a clear substrate dependence on the accessible rate of H₂O₂ production. If this method was adopted, presumably the user would have two options i) use a substrate as a fuel purely for H₂O₂ production, or ii) incorporate a desirable oxidation for a second marketable product. Substrate by-products would also need to be considered - CO₂ production from formic acid, for example, negates the zero-carbon status of electrochemical H₂O₂ production. Could the authors comment on the options for substrates for practical applications?

Author Response to Reviewers' Comments

We thank the reviewers for the constructive comments, which have helped us to greatly improve our research and the quality of our manuscript. We have now included additional analysis and performed experiments to fully address the reviewers' concerns and suggestions. Furthermore, we have modified our manuscript and SI based on the additional results and analysis, which are highlighted in yellow. Below, we address the points raised by reviewers one by one.

Reviewer #1:

This work synthesized a single-atom Rh catalyst (Rh₁/NC) to mimic the properties of flavoenzymes for the synthesis of H₂O₂ under mild conditions. The results indicated that Rh₁/NC can dehydrogenate various substrates and catalyse the reduction of O₂ to H₂O₂. The maximum H₂O₂ production rate is 0.48 mol g_{catalyst}⁻¹ h⁻¹ in the H₃PO₃ aerobic oxidation reaction. The results are very interesting, the manuscript is well-organized and easier to follow. I recommend it to be published in NC after a minor revision.

Reply: We thank the reviewer for the encouraging comment and suggestion.

1. In the Fig. 3c (in the case of benzyl alcohol solution), the activity increases with the increasing of the solution pH, however, in Fig. 4b (in the case of H₃PO₃ and HCOOH), the activity decreases with the increasing of the solution pH when the solution pH higher than 4. More discussion is needed.

Reply: We thank the reviewer for this suggestion. The first step of substrate oxidation is the breakage of the O-H bond in the -OH group. As shown in the figure below, the pKa value of benzyl alcohol is very high, indicating that benzyl alcohol can be dissociated in large quantities only under very high pH conditions. The dissociation of benzyl alcohol is highly dependent on the hydroxyl ion in the solution, so the oxidation rate of benzyl alcohol is faster under alkaline conditions. The pKa values of HCOOH and H₃PO₃ are very low, indicating that HCOOH and H₃PO₃ can be dissociated in large quantities under acidic conditions. When the pH is lower than 3.5, increasing the pH contributes to the dissociation of the O-H bond in the -OH group, so the reaction rate increases. When the pH is higher than 3.5, a large amount of HCOOH and H₃PO₃ dissociate, so the activity will not be further improved. In contrast, excessive HCOO⁻ ions, H₂PO₃⁻ ions and OH⁻ ions will be strongly adsorbed on the catalyst site, resulting in a decrease in activity (*J. Am. Chem. Soc.* 2013, 135, 9991-9994). We have added this discussion to the revised manuscript.

Figure s26. The breakage of the O-H bond in $\text{C}_7\text{H}_8\text{O}$, HCOOH and H_3PO_3 .

2. Line 213, It may be clearer if “-OH group dissociation” is revised to “the breakage of the O-H bond in -OH group”.

Reply: We sincerely appreciate this suggestion. We have corrected the improper terms in the revised manuscript.

3. line 217, the “(Figure 3d)” should be “(Figure 4d)”?

Reply: Yes, “(Figure 3d)” should be “(Figure 4d)”. We are very grateful to you for helping us find our mistakes. We have corrected the improper terms in the revised manuscript.

4. The Pt in the commercial Pt/C is Pt single atom or Pt nanoparticles? If it is Pt particles, the size can be indicated.

Reply: The Pt in the commercial Pt/C is Pt nanoparticles with a size of ca. 3 nm. We have added this information to the revised manuscript.

Reviewer #2:

In this manuscript, a single-atom rhodium catalyst (Rh_1/NC) to catalyze the reduction of O_2 to H_2O_2 in the phosphorous acid aerobic oxidation reaction, the maximum H_2O_2 production rate is $0.48 \text{ mol g}_{\text{catalyst}}^{-1} \text{ h}^{-1}$ and the selectivity of O_2 reduction to H_2O_2 can reach 100%, is reported. Although there is some depth in the review, especially in the characterization of single-atom rhodium materials. However, according to previous studies, precious metal Rh-based material can catalyze O_2 reduction to H_2O_2 . In this paper, C_3N_4 is used as N-doped carbon-supported to prevent the aggregation of Rh atoms, thus improving the efficiency of rhodium catalyzed of O_2 reduction to H_2O_2 . So how does metal Rh- C_3N_4 carrier interact with each other?

Reply: The large amount of N atoms in C_3N_4 can coordinate with the metal, which is conducive to the dispersion of Rh atoms. There are many types of N atoms in C_3N_4 , so there are various possible forms of Rh-N structures. Based on previous reports, we have summarized several possible structures, as shown in the figure below (*J. Am. Chem. Soc.* 2017, 139, 3336-3339, *Nat Commun* **12**, 6022 (2021)).

Figure s2. The coordination structure of g- C_3N_4 and Rh. Rh, red; N, yellow; C, gray.

Does another nitrogen-rich porous carbon carrier material have better catalytic efficiency? It is worth further investigation.

Reply: There are many ways to synthesize carbon-supported single-atom catalysts. The most typical synthesis method is using ZIF-8 as the carbon source and nitrogen source. We compared the activity of the catalysts synthesized by the two methods and found no obvious difference. However, the ZIF-8 method requires multiple operating steps, including hydrothermal reaction, washing, separation, and drying. Relatively speaking, our synthetic method is simpler. The precursor can be obtained by heating and melting urea and mixing it evenly with PEG and RhCl_3 . Therefore, we proposed this method and used it to synthesize a variety of single-atom catalysts.

Additionally, considering the highlight or innovation of this manuscript is also not attractive. I think this paper is insufficient to be published in Nature Communications. Several recommendations are listed as follows:

Reply: In the field of H_2O_2 synthesis, the current synthesis methods mainly focus on

electrocatalysis, photocatalysis and direct hydrogen oxidation. In the electrochemical synthesis process, many catalysts have to work in alkaline electrolytes to reduce the overpotential of the ORR. However, the low oxidability and easy decomposition of the produced H_2O_2 under alkaline conditions hindered the subsequent application for oxidation reactions. The photocatalytic method is also not suitable for the mass production of H_2O_2 due to the low efficiency (Table s3) and instability of H_2O_2 under illumination. The method of direct oxidation of hydrogen has the risk of explosion. In addition, the above three methods require complex catalytic equipments, including electrochemical workstations and H-type electrolytic cells used in electrocatalysis, light sources used in photocatalysis, and high-pressure reactors used for hydrogen oxidation.

In this work, we present an interesting report for H_2O_2 production under mild conditions by mimicking the flavoenzyme. We have summarized several innovations of the article and listed them below.

Innovations:

1. Simple method for single-atom catalysts synthesis. We proposed an innovative method to synthesize single-atom catalysts using melted urea as the solvent. Urea is used not only as a solvent to dissolve and disperse Rh and PEG but also as an N source doped on the carbon substrate to form an N-coordinated single-atom catalyst. The synthesis process does not require additional water or other organic solvents. Therefore, this synthetic method is not only simple but also more economical. In addition, this method is universal and can be used to synthesize a variety of single-atom catalysts.

2. Simple method for H_2O_2 synthesis. Single-atom catalysts can effectively catalyze the oxidation of various substrates to a target product and produce H_2O_2 . The whole catalytic reaction can be carried out in a simple container without the complex equipment required in electrocatalysis, photocatalysis and hydrogen oxidation. In addition, the H_2O_2 production rate of Rh_1/NC is 3 orders of magnitude higher than that of the photocatalysts.

3. Kinetically restrained oxygen reduction reaction. In addition to the above important innovations, we believe that the most important contribution of this work is that we found that the oxygen reduction kinetics through electrocatalysis are very different from those of enzyme mimics. In the electrocatalytic ORR, the H_2O_2 selectivity depends on the intrinsic properties of the electrocatalysts. In the enzymatic ORR, O_2 reduction is only activated under the premise of catalytic dehydrogenation of the substrate, while a single catalytic site of catalysts can only catalyze the removal of 2 electrons per substrate molecule. Therefore, the subsequent O_2 reduction is kinetically restrained to H_2O_2 through the typical 2-electron pathway with very high selectivity.

In particular, we found that the H_2O_2 selectivity in commercial Pt/C-catalyzed enzymatic reactions can reach 75%, which is 30 times higher than that in electrocatalytic O_2 reduction reactions. We think this is a shocking discovery because Pt/C is considered to be a very typical 4-electron oxygen reduction catalyst in electrocatalysis.

Based on the above innovations, we believe this work will attract extensive attention.

1. C_3N_4 was synthesized by carbonization of urea and PEG at high temperature. How was the structure of C_3N_4 determined and what was the yield? The content of Rh in RH-NC materials

needs to be revealed by more accurate characterization.

Reply: C_3N_4 is only the intermediate product of the synthesis process, not the final product. The synthesis process of catalysts only requires two steps: the first step is mixing, and the second step is pyrolysis. As shown in figure s2, we take samples at different temperatures to study the structure of materials at the synthesis stages. Urea is thermally polymerized to C_3N_4 at 500 °C to 650 °C. We found that C_3N_4 volatilized completely when heated further. Therefore, the single-atom catalyst cannot be obtained by calcining the mixture of urea + $RhCl_3$. Calcining the mixture of PEG + $RhCl_3$ will form carbon-supported Rh nanoparticles. For urea + PEG + $RhCl_3$, urea can not only promote the dispersion of Rh atoms but also facilitate the dispersion of PEG, resulting in the final formation of a single-atom catalyst with a high specific second area, which has been proven by the AC-HRTEM images, XANES, and BET tests.

Figure s3. Photos of the different mixtures after pyrolysis at different temperatures.

The elemental composition of the catalysts was quantified by ICP and elemental analysis.

Table s1. Elemental composition of catalysts.				
	C (wt%)	N (wt%)	O (wt%)	Metal (wt%)
Rh_1/NC	86.2	10.1	2.2	0.89
Ir_1/NC	85.4	10.4	2.6	1.62
Co_1/NC	88.2	9.4	1.9	0.54

2. Then the interaction between metal $Rh-C_3N_4$ carrier needs to be revealed. What is the advantage of using C_3N_4 ? During the synthesis of Rh_1/NC , “the mixture solution was... 2 °C/min”, whether the process is safe.

Reply: Figure s shows the interaction between metal $Rh-C_3N_4$ carriers.

Figure s2. The coordination structure of g-C₃N₄ and Rh. Rh, red; N, yellow; C, gray.

To be precise, we use urea to synthesize single-atom catalysts. Urea is used not only as a solvent to dissolve and disperse Rh and PEG but also as an N source doped on the carbon substrate to form an N-coordinated single-atom catalyst. The synthesis process of the catalyst only requires two steps: the first step is mixing urea + PEG + RhCl₃, and the second step is pyrolysis of the mixture. The synthesis process does not require additional water or other organic solvents. Therefore, this synthetic method is not only simple but also more economical.

Heating urea in a tubular furnace is often used in the synthesis of N-doped materials. The synthesis process is very safe. What needs to be avoided in the experiment is the blockage of the outlet because part of the urea will volatilize after heating and recrystallize into a solid at the outlet.

3. What does the standard card correspond to that should be shown in the XRD patterns (Figure 2(g)). The size dimension of all SEM or TEM images should be visualized. The tables used should be presented in a three-line format. Since the rotating ring-disk electrode of N=0.44 is used in this paper, the size of its needs to be clear.

Reply: We are very grateful to you for helping us find formatting issues. The scale bar and the corresponding value were added to the SEM or TEM images (Figure 1 and Figure 2). The tables were revised in a three-line format consistent with the format required by Nature communications. The diameter of the working electrode is 4 mm. The inner diameter of the ring electrode is 5 mm, and the outer diameter is 7 mm. We have added this information to the revised Methods section. However, it should be pointed out that the N=0.44 used here corresponds to the current rather than the current density. Therefore, the electrode area does not need to be considered in the calculation process.

Figure 2h. XRD patterns of Rh₁/NC and Rh/C.

4. Ru/NC synthesized by urea and PEG with RuCl₃ at other ratios and different temperatures should also be explored.

Reply: We thank the reviewer for this suggestion. In the early stage of this work, our synthesis conditions were optimized after extensive experiments, including different material ratios and calcination temperatures. We fixed the amount of RhCl₃ and changed the amount of urea and PEG to optimize the synthesis conditions. We added the activity test of the control materials in the revised manuscript.

Reducing the amount of urea and PEG will make Rh easy to aggregate, thus reducing the catalytic activity. Increasing the amount of urea and PEG reduced the mass fraction of Rh in the catalyst, so the mass activity was also reduced. In addition, we studied the effect of temperature on the activity of the catalyst and found that the optimal temperature was 900 °C.

Figure s19. Activity comparison of catalysts obtained under different synthesis conditions. (a) Urea, 5 g; RhCl₃, 0.005 mmol; 900 °C. (b) PEG, 200 mg; RhCl₃, 0.005 mmol; 900 °C. (c) Urea, 5 g; RhCl₃, 0.005 mmol; PEG, 200 mg.

5. As the format, the typeface in this manuscript is not unified, such as Page 11, line 325 and Page 12, line 335, and the references need to be carefully formatted, such as:5, 5, 9, 10, 11.... Please check the PDF file before submitting your manuscript.

Reply: We are very grateful to you for helping us find formatting issues. We have unified the format in the revised manuscript.

Reviewer #3:

The authors present an interesting report into a flavoenzyme-mimicking single atom catalyst for H₂O₂ production. The presentation of a distinct electron transfer mechanism versus traditional electrocatalysts is convincing and well supported by computational and experimental data. It is interesting to see the same concept being applied to multiple metals and substrates within the same report as well. I have a couple of minor comments, mostly focused on how this catalyst could be employed for wider scale production of H₂O₂. Other than these I am happy to recommend this article for publication.

Reply: We thank the reviewer for the encouraging comment and suggestion. We are happy to share our opinions in the field of H₂O₂ synthesis.

At present, the wider-scale production of H₂O₂ relies on the well-established anthraquinone process in industry. The anthraquinone method is predominantly used and accounts for more than 95% of H₂O₂ production. However, this centralized production is energy intensive and involves purified H₂ and concentrated aqueous H₂O₂, which introduce safety risks in storage and transport. To minimize transportation costs, energy-intensive distillation is required to produce up to 70 wt % H₂O₂. Nonetheless, there are multiple applications where end-users require dilute H₂O₂ solutions. For water treatment, only <0.1 wt % H₂O₂ is necessary. Highly concentrated H₂O₂ is explosive, and its transportation has caused severe accidents (ACS Catal. 2018, 8, 4064-4081).

Since dilute H₂O₂ is more suitable for a wide range of applications, researchers have proposed electrochemical and photocatalytic methods to synthesize H₂O₂ in an attempt to reduce costs and risks. The purpose of developing these synthetic methods is to realize the on situ synthesis of H₂O₂ instead of replacing the industrial method to synthesize H₂O₂. These goals also indicate that on-site synthesis methods are difficult to achieve on an industrial scale.

It is worth pointing out that the H₂O₂ production by aerobic oxidation of formic acid does not require complicated equipment (only need a beaker), so it is easier to achieve large-scale production to a certain extent compared with electrocatalytic and photocatalytic methods.

I may have missed this, but I couldn't see a reference to the experimental setup for the enzymatic H₂O₂ production. Factors such as mass loading of Rh/NC, dispersion method, if it was stirred during reactions etc.

Reply: We thank the reviewer for this suggestion. The missing information has been added to the revised experimental section.

H₂O₂ production and quantification.

For H₂O₂ production, 20 μL Rh₁/NC (0.5 mg mL⁻¹, dispersed by ultrasound) and 20 μL substrates (125 mM H₃PO₃ + 125 mM KH₂PO₃) were added into a vial containing 920 μL H₂O. After the solution was mixed, the reaction was carried out without stirring for several minutes at room temperature. The detailed reaction conditions (concentration and reaction time) of different experiments were added to the corresponding figure notes.

For H₂O₂ quantification, after the reaction of catalysts and different substrates, 20 μ L HRP (0.1 mg mL⁻¹) and 20 μ L TMB (20 mM in DMSO:EtOH=1:9) were added to the above solution (960 μ L). UV–vis absorption measurements were performed within 2 min. The produced H₂O₂ was quantified by a standard curve, which was drawn from a series of known concentrations of H₂O₂.

Have the authors considered how the Rh/NC could be used for lab-scale or industrial-scale H₂O₂ production? Electrocatalysts have the advantage of existing reactor designs, and flow-by electrochemical cells can remove H₂O₂ as it is produced into separate storage containers. Presumably the Rh/NC would need a separate process of H₂O₂ extraction and Rh/NC recovery for batch production?

Reply: The catalyst can be recovered by centrifugation. The H₂O₂ produced can be more effectively used in oxidation reactions under weakly acidic conditions. As discussed earlier, because the synthesis method is very simple, this method more easily achieves large-scale production to a certain extent. For example, the catalyst can be loaded on common commercial supports, such as aluminum oxide or titanium dioxide, and applied to a fluidized bed reactor to realize large-scale production.

Compared with the separation of catalysts, we believe that the separation, storage, or utilization of H₂O₂ is more important. H₂O₂ is synthesized because it is a green strong oxidant and can be used in a variety of oxidation reactions. In electrocatalysis, although H₂O₂ can be easily separated from the catalyst, it is difficult to separate from the electrolyte. The suitable pH for H₂O₂ to participate in the oxidation reaction is usually in the range of 3-5, while the common electrolyte is a strong acid or strong base (Table s3), which is not conducive to the direct utilization of the on-site synthesized H₂O₂. In addition, H₂O₂ not only has a weak oxidizing ability but also easily decomposes under alkaline conditions, so it is difficult to find practical applications for the synthesis of H₂O₂ under alkaline conditions. In our method, the catalyst can be separate by centrifugation. The pH value of the reaction solution is approximately 3, which is very beneficial for the H₂O₂ to be used in the subsequent oxidation reaction. Therefore, our synthetic method is very promising for application to practical reactions, including hydrocarbon bond oxidation, sterilization, pollutant degradation, and bleaching.

Table s3. Comparing the performance of Rh ₁ /NC and electrocatalysts for O ₂ reduction to H ₂ O ₂ .				
Catalyst	pH	Productivity (mmol g _{cat} ⁻¹ h ⁻¹)	Selectivity (%)	Reference
Rh ₁ /NC	3	480	~100	This work
Pt-Hg/C	1	/	96	Nat. Mater. 2013 , 12, 1137
O-CNTs	13	111.7	~90	Nat. Catal. 2018 , 1, 156.
F-mrGO	13	430	~100	Nat. Catal. 2018 , 1, 282

Pt ₁ /HSC	1	/	96	Nat. Commun. 2016 , 7, 10922
g-N-CNHS	13	/	63	Chem 2018 , 4, 106
Pt ₁ -CuS _x	1	546	~95	Chem 2019 , 5, 2099
Au-Pd	1	/	~95	J. Am. Chem. Soc. 2011 , 133, 19432
Au-Pt-Ni	13	/	~95	Adv. Mater. 2016 , 28, 9949-9955
Co ₁ -NG(O)	13	418	~80	Nat. Mater. 2020 , 19, 436
Co SAC	1	80	~90	Chem 2020 , 6, 658
Co-POC-O	13	/	~85	Adv. Mater. 2019 , 31, 1808173
Fe-CNT	14	~1600	~90	Nat. Commun. 2019 , 10, 3997
Co-N-C	13	193	~70	J. Am. Chem. Soc. 2019 , 141, 12372

Similarly, the authors show a clear substrate dependence on the accessible rate of H₂O₂ production. If this method was adopted, presumably the user would have two options i) use a substrate as a fuel purely for H₂O₂ production, or ii) incorporate a desirable oxidation for a second marketable product. Substrate by-products would also need to be considered - CO₂ production from formic acid, for example, negates the zero-carbon status of electrochemical H₂O₂ production. Could the authors comment on the options for substrates for practical applications?

Reply: We are very grateful to the reviewer for helping us point out the innovations of this work. Indeed, the single-atom catalysts can not only effectively catalyze the oxidation of various substrates to a target product but also catalyze the reduction of O₂ to H₂O₂. Therefore, the Rh₁/NC catalyst and this kind of catalytic reaction will be widely used.

Although formic acid will produce carbon dioxide after oxidation, formic acid can still be considered a green molecule. As shown in the following two examples, HCOOH has been proposed as a convenient hydrogen storage material that contains 4.4 wt% hydrogens; a direct formic acid fuel cell uses formic acid as fuel to provide energy output.

(Decomposition of formic acid for hydrogen production)

(reaction in a formic acid fuel cell)

In addition, for our reaction system, the generated CO₂ will form H₂CO₃ after dissolving in water (acidic), which is conducive to the preservation and utilization of H₂O₂.

To synthesize H₂O₂, formic acid and phosphite can be used as substrates for the rapid synthesis of H₂O₂.

Rh can oxidize a variety of substrates, so it is also widely used to synthesize specific products or to consume unwanted molecules. For example, in cancer treatment, glutathione has antioxidant properties, which reduce the ability of reactive oxygen species to kill cancer cells. The Rh₁/NC catalyst can not only catalyze the oxidation of glutathione but also produce H₂O₂, so it is promising for use in killing cancer cells. This is also a good example to illustrate

the advantages of enzymatic catalysis over electrocatalysis and photocatalysis to synthesize H_2O_2 in situ.

REVIEWERS' COMMENTS

Reviewer #1 (Remarks to the Author):

The authors have answered the questions raised by the reviewers and revised following the comments. My suggestion is therefore to accept this manuscript for publishing in NC as is.

Reviewer #2 (Remarks to the Author):

The authors have responded my comments well. After revision, I think it can be accepted.

Reviewer #3 (Remarks to the Author):

The manuscript was of a high quality before any comments were made. The comments raised in my review have been addressed fully and in a good amount of detail. The authors should be commended on a high quality report. I am happy to recommend this for publication.